# Bioclimatic Characterisation of Specific Native Californian Pinales and Their Future Suitability under Climate Change

**DOI:** 10.3390/plants12101966

**Published:** 2023-05-12

**Authors:** Alejandro González-Pérez, Ramón Álvarez-Esteban, Ángel Penas, Sara del Río

**Affiliations:** 1Department of Biodiversity and Environmental Management (Botany Area), Faculty of Biological and Environmental Sciences, University of Leon, Campus de Vegazana s/n, 24071 León, Spain; 2Department of Economics and Statistics (Statistics and Operations Research Area), Faculty of Economics and Business, University of Leon, Campus de Vegazana s/n, 24071 León, Spain; ramon.alvarez@unileon.es; 3Department of Biodiversity and Environmental Management (Botany Area), Faculty of Biological and Environmental Sciences, University of Leon, Mountain Livestock Institute CSIC-UNILEON, Campus de Vegazana s/n, 24071 León, Spain; apenm@unileon.es (Á.P.); sriog@unileon.es (S.d.R.)

**Keywords:** bioclimatology, climate change, California conifers, habitat suitability, species range shift

## Abstract

Rising temperatures and changes in precipitation patterns under climate change scenarios are accelerating the depletion of soil moisture and increasing the risk of drought, disrupting the conditions that many plant species need to survive. This study aims to establish the bioclimatic characterisation, both qualitative and quantitative, of ten native Californian Pinales for the period 1980–2019, and to determine their habitat suitability by 2050. To achieve this, an exhaustive search of the Gbif database for records of ten conifer taxa was carried out. To conduct the bioclimatic characterisation of the studied taxa, we worked with the monthly values of average temperature and precipitation for the period 1980–2019 from 177 meteorological stations. Linear regressions was performed in order to compile the future evolution of California’s climate. Suitable areas and optimal areas were defined at the present time (1980–2019) and its future projection (2050). We applied Boolean logic and, in this investigation, the Conditional Logic Operator (CON) was used to determine the possible species presence (one) or absence (zero) for each of the 15 variables analysed. In general, most of the conifers studied here will experience a reduction in their habitat range in California by the year 2050 due to climate change, as well as the displacement of species towards optimal areas. Furthermore, the results have highlighted the applicability of bioclimatology to future conditions under climate change. This will aid conservation managers in implementing strategic measures to ameliorate the detrimental impacts of climate change, thereby ensuring the ecological integrity and sustainability of the affected conifer species.

## 1. Introduction

The Intergovernmental Panel on Climate Change (IPCC) reports that the Earth’s temperature is expected to increase by 1.5 °C above pre-industrial levels by 2050 [1,2]. This temperature increase will cause the sea level to rise, enlarge the frequency and severity of extreme weather events, and cause irreversible damage to ecosystems [3]. Climate change is affecting California’s biodiversity [4,5] by altering temperature [6,7,8,9], precipitation [10,11,12,13], and weather patterns [14,15]. These changes can alter the timing of seasonal events, such as flowering and migration, and cause shifts in species distribution ranges [16,17,18].

Loss of vegetation is a major environmental concern in California, as it can have a significant impact on the state’s ecosystems and the services they provide. In fact, a more diverse plant community can provide a wider range of ecosystem services [19]. Vegetation die-off can be caused by a variety of factors, including urbanisation, forest exploitation, forest fires, and climate change [3,20,21,22,23,24].

One of the major causes of vegetation loss in California is urban development. As the state’s population has grown, more land has been devoted to residential, commercial, and industrial uses. This has led to the destruction of natural habitats and the shortfall of native vegetation [25,26]. Urban expansion can have a negative impact on vegetation by creating so-called “heat islands” [27]. These are areas within a city that are significantly warmer than their surroundings, caused by the high number of buildings and paved surfaces. Heat islands can cause harm to both animals and plants, leading to a loss of vegetation [28]. This effect is particularly pronounced in urban areas where there is a lack of vegetation to begin with, as plants and trees play an important role in mitigating the impacts of heat islands. Moreover, climate change has a significant impact on the distribution and abundance of tree species [29]. Rising temperatures and changing precipitation patterns [6,10] are accelerating soil water depletion and increasing the risk of drought, altering the conditions that many plant species need to survive [30]. These changes are often pronounced in ecotones where species reach their climatic bounds and are replaced by other species [31]. In addition, habitat loss affects not only the trees, but also a wide variety of wildlife [29]. The effects of global warming will continue to wreak havoc on water availability on the west coast of the U.S. [32].

Plants and some tree species in particular may not be able to adapt quickly enough to changing conditions [33,34]. Drought has been the cause of forest mortality events in the state of California, often in combination with other abiotic and biotic factors [34,35,36,37,38]. Furthermore, vulnerability to drought is of fundamental importance in determining the geographic distribution of individual species and communities. Recent research suggests that rising global temperatures and precipitation disturbances are already reinforcing drought-induced forest conversion [39,40,41,42,43]. In essence, warming would lead to higher episodic forest mortality at lower elevations and lower runoff generation at higher altitudes [44,45]. Several studies have shown a huge loss of snowpack in the Sierra Nevada due to earlier melting and more rain instead of snow [46,47]. The future consequences of longer droughts and higher temperatures are potentially drastic.

To avoid this situation, an approach such as bioclimatology could be a potential solution. Bioclimatology is an ecological science that studies the relationships between climatic conditions and the distribution of living beings and their communities around the world [48,49,50]. Other authors have also studied changes in the habitat suitability of some plants using other methodologies [33,51,52].

According to the Worldwide Bioclimatic Classification System by Rivas-Martínez et al. (2011), bioclimatic units are precisely correlated with vegetation models and climate values. The increase in comprehensive knowledge of the distribution of vegetation on Earth, as well as changes in the form and composition of potential vegetation, has allowed scientists to identify bioclimatic thresholds to define them. It is noteworthy that bioclimatic models can predict the future responses of plants to climate change, focusing on climatic thresholds of species distribution [53].

California was a prolific timber producer in the 20th century, and many of the conifers are of great economic importance today [54]. Giant redwoods have imposing massive trunks that are of great importance to the tourism industry. They have made California’s majestic forests famous all over the world [55]. The coniferous forests of California became increasingly interesting both culturally and scientifically.

California’s Mediterranean biome is a unique and important ecosystem that owes its existence to the novel proto-Mediterranean climate and fire regime of the middle Miocene, about 15 million years ago [56]. The result is a diverse flora that is dominated by radiations of herbaceous and sub-shrub lineages in families such as Asteraceae, Boraginaceae, and Lamiaceae, as well as many endemic species that have evolved to thrive in the Mediterranean climate [56]. California is a hotspot for conifer diversity, with 58 species within the state, including notable ancient paleoendemic species such as *Sequoia sempervirens* and *Sequoiadendron giganteum* [57,58]. Of the conifer species in California, 31 are endemic to the region, including forest dominants and others that compete successfully with angiosperms on oligotrophic or azonal soils. This represents 9% of all conifer species in the world [59].

Nevertheless, the frequency and intensity of droughts have increased in recent years, leading to reductions in the size and vigour of some conifers [33,60]. Collectively, the species studied grow in the Cascade ranges, the North and Central, the Sierra Nevada, the Intermountain, and the Southwestern California Provinces [61]. Some of these have restricted ranges at high elevations (the Sierra Nevada and the Cascade Ranges Province), so changes in the magnitude or timing of snowfall and snowmelt in these areas will affect water availability, which could lead to upslope contraction of conifer forests in scenarios of lower water availability, or downslope expansion in scenarios of higher water availability [62]. Others are specific to the Pacific Northwest region of the United States and Canada. In the context of climate change, these areas may be reduced, as is the case for other native Californian species [63]; indeed, the range and survival of these trees are threatened by the effects of global warming in the state of California [31,64,65,66,67,68,69]. In addition, the increase in wildfires due to global warming is another threat that this species may have to cope with [70,71]. The taxa selected for this research, whose economic, cultural, and scientific usefulness is invaluable, may be more vulnerable, and to a greater extent, to the effects of climate change.

To anticipate the potential effects that climate change may have on conifers in California, this research has the following main objectives:

To establish both the qualitative and quantitative bioclimatic characterisation of ten native Pinales Gorozh for the period 1980–2019. The taxa analysed are: Abies procera Rehder; Abies magnifica A. Murray bis; Pinus longaeva D.K. Bailey; Pinus jeffreyi A. Murray bis; Pinus balfouriana A. Murray bis; Pinus albicaulis Engelm; Picea sitchensis (Bong.) Trautv. & C.A. Mey; Tsuga mertensiana (Bong.) Sarg; Sequoiadendron giganteum (Lindl.) J. Buchholz; Sequoia sempervirens Endl.

To determine the suitable and optimal areas for the analysed taxa under current climate conditions and to analyse if there are zones of gain, loss, and maintenance of habitat suitability under future climate scenarios (2050).

Former research has used Thornthwaite methodology to describe the distribution of tree species in Yosemite Park [33]. However, this paper goes further and, to the best of our knowledge, is the first to analyse the whole state of California applying the Worldwide Bioclimatic Classification System methodology [48]. In this respect, the quantitative bioclimatic characterisation is accomplished as a novelty. Lastly, it is noteworthy that very little research has been conducted on the impact of climate change on the distribution of the taxa studied.

The assessment and prediction of bioclimatic parameters and indices that characterise and define plant communities could provide valuable information for planners and policy makers to deal with the consequences of climate change [72]. In this way, this research can help environmental managers throughout their local area [73] to take strategic measures to mitigate and anticipate the adverse effects of global warming on the loss of habitat suitability for the conifer species studied.

## 2. Materials and Methods

### 2.1. Study Area

California is located on the west coast of the United States. It covers an area of 423,955 km^2^ and is one of the most populated states in the country. It is bordered by Oregon to the north, Nevada and Arizona to the east, and the Mexican State of Baja California to the south.

The state’s highest peak (4421 m) is Mount Whitney in the Sierra Nevada. This mountain range also includes Yosemite National Park, famous for its waterfalls, rock formations, vegetation, and wildlife. This territory is home to many national parks, including Yosemite National Park [74], Death Valley National Park, and Sequoia National Park.

The Central Valley is a large agricultural region that runs through the centre of the state. It is host to the Sacramento Valley, whose rivers are the state’s main source of water [75]. California has a complex water system, with many rivers and reservoirs providing water for irrigation, hydroelectric power, and drinking water. The state is divided into two main water systems: the Colorado River Aqueduct and the State Water Project [76,77]. The Central Valley is also home to many fruit and nut trees, such as almond, pistachio, and orange trees. Agriculture is a major industry in California [78,79], producing a wide variety of crops such as grapes, almonds, and lettuce. In addition, the state is also home to many wineries and is a leading producer of wine [80].

The eastern part of the state is home to the Mojave Desert and Death Valley, which is the lowest and hottest point in the United States. The two southern desert zones, the Mojave and the Sonoran (Colorado) are part of the Basin and Range geomorphic province [81]. The state is also home to several active and latent volcanoes, including Mount Lassen and Mount Shasta [82]. As noted above, the state has a diverse landscape, including sandy beaches, redwood forests, the Sierra Nevada mountain ranges, and arid desert regions. As a result, a wide variety of vegetation displays over this territory. Some of the major vegetation types listed by Barbour et al. (2007) [81] are: chaparral, desert communities, grasslands, wetlands, redwood forest, coniferous forest, evergreen forest, and mixed forest.

From a bioclimatic point of view, California is divided into three macrobioclimates [48]. Tropical is found in the Sonoran Desert. Mediterranean covers more than three-quarters of its surface. This macrobioclimate is characterised by *p* < 2T, where P is the average rainfall (in millimetres) of the two hottest months of the summer quarter, and T is their average temperature expressed in centigrade. This condition applies to two consecutive months. There is also a Temperate macrobioclimate in the Klamath highlands. From a biogeographical perspective, California is made up of 9 biogeographical provinces [61], which from north to south are as follows: The Cascade Ranges, located on the north coast, together with the North and Central California Province and, in the extreme northeast, the Columbia Plateau Province. Further into the central part of the state and to the east are the Central Valley Province, the Sierra Nevada Province, and Intermountain Province. In the south are the Southwestern California, Mojave Desert, and Sonora Desert Provinces.

### 2.2. Conifers’ Occurrence Data

Due to the effects of drought on reducing the size and vigour of some conifers, and the fact that they have a limited range at high altitudes, we have selected for this research taxa with economic, cultural, and scientific value that may experience the impact of climate change to a greater extent.

Accordingly, ten native taxa of arboreal gymnosperms were selected to achieve one of the main objectives of the work, already mentioned in the introduction.

A thorough search of these taxa was carried out on the Gbif website (https://www.gbif.org/es/) (accessed on 10 January 2023). For each one of the ten taxa mentioned above, we searched for points of presence, with a total of 7000 occurrences. From these points, we collected the species name, coordinates (latitude and longitude), elevation, and presence in the state of California. We then compared them with the proposed distribution in the Calflora database (https://www.calflora.org/) (accessed on 10 January 2023), which is widely recognised as one of the most authoritative sources of information on California plants, and is frequently used as a reference by academic researchers, conservationists, and land managers. We then eliminated those that were repetitive, were of dubious provenance, or lacking a precise locality. This left us with a total of 3506 occurrences (Figure 1).

### 2.3. Conifers’ Descriptions

According to the Catalogue of life (https://www.catalogueoflife.org/) (accessed on 30 April 2023), the conifer species studied belong to the families *Pinaceae* Spreng. ex F.Rudolphi and *Cupressaceae* Gray. They are also all members of the order Pinales.

#### 2.3.1. *Abies magnifica*

The red fir dominates the upland areas of California and has been a forest species of great importance in the state, as the wood is used in general construction [54]. It generally lives in a cold, moist environment with abundant precipitation (556.2–929.9 mm) in the form of snow. The soils on which it grows are young and in the order of Entisols, Inceptisols, and Alfisols [83]. This tree is a dominant species in most of the territory, sharing vegetation cover with *Abies concolor* Lindl. & Gordon, *Pinus contorta* Bol. and *Pinus jeffreyi* A. Murray. In the Shasta Mountains, it is found with *Abies procera* Rehder and *Pinus balfouriana* A. Murray bis [84].

#### 2.3.2. *Abies procera*

Noble fir, also known as red fir, is a fir of impressive size, and its habitat is limited to the Cascade Range. In California, it occurs in disjunct southern areas of this mountain system, with great morphological variability [81,84]. This tree can also grow on high moisture soils and is most common on Spodosols and Inceptisols [54,85]. This type of forest is associated with most of the conifers in the northern part of the state, and can be found in the western Shasta Mountains with *Abies amabilis* J. Forbes and in the east with *Picea sitchensis* (Bong.) Trautv. & C. A. Mey [84]. It has a high strength−to−weight ratio and has been used as an aircraft construction material [54].

#### 2.3.3. *Picea sitchensis*

Sitka spruce grows in a narrow strip along the northern Pacific coast. Its range in California is being limited to the northwest coast of the state [54]. The climatology is typical of coastal areas with abundant continuous moisture, mild winters, and cool summers. Sitka spruce flourishes in Entisols, Spodosols, Inceptisols, and Histosols but prefers deep, moist, well−aerated soils [83].

#### 2.3.4. *Pinus albicaulis*

Whitebark pine is valued for watershed protection and aesthetics, as well as for its seeds, which provide food for brown bears, among others. This species has been listed as endangered on the IUCN red list of Threatened Species since 2011 [86]. This sapling grows in higher elevation forests above the subalpine ring to the highest peaks of the Klamath Mountains. It thrives in areas with a climatology characterised by cold, windy, and wet conditions, with an average annual precipitation of 600–1800 mm, mostly in the form of snow [54]. This type of forest is found on poorly developed soils, often classified as Inceptisols, with a slightly acid pH.

#### 2.3.5. *Pinus balfouriana*

A tree known as foxtail is found on upper slopes, ridges, and peaks up to the forest line. It has as a special use, the craftsmanship, for dead or fallen wood since most stands of this species are found in protected areas. The soils on which it grows are loose, well−drained, and often rocky, even glacial. It is located in several areas throughout the region, although there are two distinct disjunct zones that may have originated 8000 years ago occupying areas of ancient glaciers. It grows on Inceptisols, serpentine soils of the Klamath, and the Sierra Nevada mountains.

#### 2.3.6. *Pinus jeffreyi*

The range of Jeffrey pine extends north through the Klamath Mountains and throughout the Sierra Nevada, south from the Transverse and Peninsular Ranges to northern Baja California [84,87]. This tree is particularly drought−tolerant and cold−hardy, and dominates other conifers with which it shares its habitat. It is associated with *Libocedrus decurrens* Torr., *Pseudotsuga menziesii* (Mirb.) Franco, in most of the territory, although it also shares some sites with *A. magnifica* [81]. The only areas where it grows exclusively are east of the Sierra Nevada in the Lake Tahoe area.

#### 2.3.7. *Pinus longaeva*

The bristlecone pine is a long−lived tree of arid and subalpine environments at high altitudes in the western U.S., reaching almost 4900 years old and 12 m in height [84,88]. It has multiple stems and is associated with *Pinus flexilis* A. Murray bis [81,84]. Both species are priorities for conservation due to their cultural and ecological importance [89].

#### 2.3.8. *Sequoia sempervirens*

The redwood or coast redwood is one of California’s iconic trees, with 95% of its range on the central and north coast of the state in Inceptisols and Ultisols [54,83,87]. It is found in coastal areas isolated from oceanic salinity and where summer fog is common [54]. This is of great importance for this species, as it has been shown that *S. sempervirens* can absorb small amounts of fog water directly through its leaves [90]. Asexual regeneration is prolific and gaps created by disturbance influence the regeneration of *S. sempervirens*. In addition, regeneration of this species is high on the edges of these ponds [91].

#### 2.3.9. *Sequoiadendron giganteum*

The well−known giant sequoia develops in the Mediterranean climate, in high areas with humidity and hot and dry summers. It shares forest cover with *Pinus lambertiana* Douglas, *Alnus rhombifolia* Nutt, and *Abies magnifica* A. Murray bis. Since it has wide public appeal and limited natural range, most groves are protected and it is listed as an endangered species by the IUCN [92]. If, as predicted, the future climate in the Sierra Nevada will lead to reduced snow and summer precipitation, this species will face increasing competitive pressures in its current range [90,93].

#### 2.3.10. *Tsuga mertensiana*

Mountain hemlock is found in cold and snowy subalpine areas where it reproduces slowly [54]. It has also been reported on organic soils (Histosols) and also on immature soils (Entisols and Inceptisols) [54,85]. It grows in moist conditions with a high humus content in the soils where this species occurs, and also forms stands on north−facing lake margins with late snow [94]. Mountain hemlock often flourishes mixed with other trees and has many associated plant species such as *A. magnifica* A. Murray bis, *Abies concolor* Lindl. & Gordon, and *Pinus albicaulis* Engelm [84,85]. The wood of mountain hemlock is commonly used for mines, interior finishes, boxes, kitchen cabinets, and floors and ceilings [95].

### 2.4. Climate Data and Bioclimatic Characterisation

Initially, we used a database of 350 weather stations with monthly mean temperature and monthly precipitation data available on the Western Regional Climate Center (WRCC) website [96]. Weather stations were selected based on criteria of completeness, location, and homogeneity across California [12]. Only stations with less than 10% missing values were chosen. These values were fulfilled with the corresponding long−term monthly mean [97]. To carry out the bioclimatic characterisation of the studied taxa, we finally worked with monthly mean temperature and precipitation values for the period 1980–2019 from 177 meteorological stations. With the rainfall and temperature data obtained, different bioclimatic parameters and indices (Table 1) were calculated following the Rivas−Martínez bioclimatic classification system [48]. The qualitative bioclimatic characterisation aims to establish, for each taxon, the isobioclimate, that is: macrobioclimate, bioclimate, bioclimatic variant, thermotype, ombrotype with their bioclimatic horizons, and continentality. The quantitative characterisation shows the minimum, interquartile range, and maximum values of each parameter and index studied in which each taxon can occur.

### 2.5. Future Climate Projections

Predicting the future climate is a complex and challenging task. There are several approaches to climate prediction, including analysis of observational data, numerical models, and statistical methods [98]. Linear trend analysis is a statistical method used to identify the existence of a linear relationship between a dependent variable and one or more independent variables [99]. The aim of linear trend analysis is to determine whether the data points in a dataset are increasing or decreasing at a consistent rate over time. This can be useful for identifying patterns in data such as bioclimatic indices [100].

Linear and non−linear regressions have been applied to solve a number of computational problems [101]. A major weakness of existing regression approaches is their lack of strength to outliers and noise. To avoid this, robust regression is often used to provide strong results in the presence of outliers, reducing the weight given to these outliers or even eliminating them. This is of particular concern when forecasting variables such as precipitation (less so for temperature). To achieve this stability, robust regression confines the influence of outliers [102,103,104,105,106,107].

We have addressed this problem by using four robust regression models. With the aim of capturing the evolution of California’s climate for the future, several methodologies were used, such as simple linear least squares regression analysis, quadratic regression, cubic, and robust regression to obtain the prediction of temperature, precipitation, and bioclimatic indices for the year 2050. The results were summarised for each season, month, and year. The quadratic and cubic models did not significantly improve the predictions for the known period of time. Forecasts from the year 2019 onwards showed rapidly increasing or decreasing values, excessive sensitivity, and predicted values that did not correspond to the limits (minima and maxima) observed for the period analysed.

Two types of Box–Jenkins time series analyses were run using the auto.arima function from forecast R package [108] ARIMA−SARIMA for each station using the standard correction to Akaike’s Information Criterion (AICc) correction to find the best model. In the same way, 2124 ARIMA models were run (177 stations × 12 months). The results of the methods used in this study will be presented in a forthcoming publication. By itself, it was not possible to present these results in this paper.

Four robust linear regression were addressed using different packages and algorithms: Theil−Sen regression [109,110] and Siegel regression method (from the RobustLinearReg R package), rlm function from the MASS R package [111] fitting a linear model via robust regression using an M estimator, and lmrob from the robustbase package [112].

To select the best model for each of the 2124 elements to be predicted (177 stations × 12 months), five regression models (including simple linear regression) were fitted to the known data. The process was repeated for both precipitation and temperature. The best results for temperature were obtained with lmrob in 53.1% of the cases, linear least squares in 46.7%, and rlm with 0.2%. For precipitation, it was lmrob with 95%, with the remaining 5% for linear regression with least squares.

### 2.6. Habitat Suitability Analysis

Individual species only thrive within certain ranges of environmental conditions [113,114]. Therefore, mapping the suitability of habitats for different plant species can help to identify areas that may be more resilient to the effects of climate change [115,116]. To achieve one of the objectives of this research, we defined suitable areas (based on the maximum and minimum values of each species for each of the variables) and optimal areas (based on the values between the first quartile (Q_1_) and third quartile (Q_3_) of each species for each of the variables) at the present time (1980–2019) and their projection into the future (2050).

With the values of the bioclimatic indices of the 177 weather stations, raster files were generated using statistical geoprocessing tools, more specifically the Empirical Bayesian Kriging (EBK) with ArcGIS 10.8 © software [117], at the finest spatial resolution possible (1 km^2^). In addition, in the habitat suitability analysis we included soil orders [83] and Digital Elevation Model (DEM) for California.

To determine habitat suitability, we applied Boolean logic that can be used in its design and internal operation, starting with an array of pixels to display the numbers. In the raster calculator, each number or symbol is displayed using a selection of pixels to form the desired image. Boolean logic is used to control the states of each pixel in the matrix, so that numbers and symbols can be showed correctly [118]. This involves the use of logical operators such as conjunction (AND), disjunction (OR), and negation (NOT). In this research, the conditional logical operator (CON) was used to establish the possible presence (1) or absence (0) of the species for each of the variables.

Once this was performed for each variable analysed, map algebra was applied using the ArcGIS 10.8 © raster calculator and Boolean operators to estimate the gain, loss, or maintenance of habitat suitability up to the year 2050 in relation to the current period.

## 3. Results and Discussion

### 3.1. Bioclimatic Characterisation of Selected Taxa

One of the aims of this research is, firstly, to carry out a quantitative and qualitative bioclimatic characterisation of the ten selected taxa. The qualitative bioclimatic characterisation was established following the bioclimatic classification of Rivas−Martínez et al. (2011). As an example (Table 2), we have indicated the most frequent isobioclimate in which a taxon grows, and brackets are used to indicate when a taxon may occur in another isobioclimate on an ad hoc basis. Quantitative diagnoses include the minimum value, interquartile range (bold and bracketed), and maximum value for each variable examined. The specific values, such as mean, are given in Appendix A, Table A1. This section contains our main results and discussion, and the abbreviation for each parameter and bioclimatic index can be found in Table 1. To summarise these results, the isobioclimates for each of the species studied are given in Table A2 in the Appendix A.

#### 3.1.1. *Abies magnifica*

This species is the characteristic of the association *Abietetum magnificae* Oosting & Billings ex Rivas−Martínez & Sánchez−Mata 1997 [119]. Macroforests dominated by this species are usually shady and dense where *Tsuga mertensiana* is common in the upper mountains [120]. Our results of ombrotype values agree with those proposed by a previous study [119], but our thermotype values differ and clarify the existence of the species in the Temperate macrobioclimate with a submediterranean variant.


**Bioclimatic characterisation**


Mediterranean pluviseasonal oceanic, Mediterranean pluviseasonal continental, Mediterranean pluviseasonal xeric oceanic, and Mediterranean pluviseasonal xeric continental semicontinental lower mesomediterranean to lower oromediterranean lower subhumid to upper humid.

(Temperate oceanic submediterranean semicontinental upper supratemperate lower hyperhumid).

**T_avg_**: 14.1 **(9.2–14.1)** 16.6; **T_min_**: −2.4 **(0.1–4.4)** 7.8; **T_max_**: 15.4 **(19.6–24.6)** 27.4; **P_avg_**: 296.1 **(669.7–1098.3)** 1994.2; **Pp**: 206.8 **(556.2–929.9)** 1567.9; **It**: 21.7 **(93.0–233.3)** 304; **Itc**: 29.1 **(103.8–257.9)** 321.1; **Tp:** 875.0 **(1134.6–1713.7)** 1996.2; **Ic:** 7.7 **(19.0–20.5)** 23.0; **Io:** 1.6 **(4.0–7.9)** 15.1; **Ios_2_:** 0.1 **(0.2–0.5)** 0.9; **Ios_3_:** 0.1 **(0.2–0.7)** 1.4; **Ios_4_:** 0.2 **(0.5–1.3)** 2.6.

#### 3.1.2. *Abies procera*

This species is the characteristic of *Tsugo mertensianae−Abietetum procerae* Rivas−Martínez, Sánchez−Mata & Costa 1999, which grows throughout the Cascade Range Sector territories [120]. Although this plant community has previously been characterised as Temperate oceanic [120], it is noteworthy that *A. procera* in California is predominantly Mediterranean in our results.


**Bioclimatic characterisation**


Mediterranean pluviseasonal oceanic euoceanic lower supramediterranean humid to lower hyperhumid.

(Temperate oceanic submediterranean weak semicontinental upper supratemperate lower hyperhumid).

**T_avg_:** 7.2 **(8.9–11.6)** 13.4; **T_min_:** −0.3 **(1.0–3.7)** 7.0; **T_max_:** 16.8 **(17.1–21.4)** 22.1; **P_avg_:** 611.9 **(1179.7–1829.8)** 1968.3; **Pp:** 633.4 **(971.7–1410.6)** 1586.8; **It:** 66.3 **(122.8–189.2)** 275.3; **Itc:** 61.7 **(127.3–196.4)** 275.9; **Tp:** 898.0 **(1091.0–1392.0)** 1609.0; **Ic:** 12.6 **(14.8–19.8)** 20.4; **Io:** 3.8 **(8.9–13.6)** 14.9; **Ios_2_:** 0.0 **(0.5–0.8)** 0.9; **Ios_3_:** 0.1 **(0.8–1.3)** 1.4; **Ios_4_:** 0.4 **(1.4–2.3)** 2.5.

#### 3.1.3. *Picea sitchensis*

This species is associated with *Sequoia sempervirens* Endl. in some parts of its distribution [81]. Sitka spruce exhibits a wide range of variation in adaptive traits and has shown a high capacity to respond to selection and adapt to past climate change along the Pacific coast of North America [121].


**Bioclimatic characterisation**


Mediterranean pluviseasonal oceanic subhyperoceanic supramediterranean humid to lower hyperhumid.

**T_avg_:** 9.2 **(9.7–11.8)** 12.5; **T_min_:** 3.8 **(4.6–7.7)** 8.1; **T_max_:** 15.4 **(15.5–15.8)** 17.2; **P_avg_:** 912.4 **(1094.3–1827.6)** 1899.2; **Pp:** 931.2 **(1082.3–1570.3)** 1793.3; **It:** 167.4 **(186.4–273.2)** 283.3; **Itc:** 159.0 **(177.7–268.0)** 279.3; **Tp:** 1119.0 **(1078.0–1418.0)** 1505.0; **Ic:** 7.8 **(8.1–10.6)** 12.2; **Io:** 6.3 **(7.7–13.5)** 14.0; **Ios_2_:** 0.2 **(0.4–0.8)** 0.9; **Ios_3_:** 0.4 **(0.7–1.4)** 1.5; **Ios_4_:** 1.0 **(1.3–2.5)** 2.6.

#### 3.1.4. *Pinus albicaulis*

This tree is associated with *Abies magnifica* A. Murray bis and locally with *Pinus balfouriana* A. Murray bis [81,84]. It is also characteristic of the association *Carici rossii−Pinetum albicaulis* D.W. in Rivas−Martínez & Sánchez−Mata 1997. This forest grows in the upper oromediterranean humid−hyperhumid zone, and it is widespread in the summit areas of the Klamath and the Sierra Nevada [119]. In some locations, our results are supported by this previous research, but our interquartile range found that *P. albicaulis* occurs in less humid areas and lower thermotypes.


**Bioclimatic characterisation**


Mediterranean pluviseasonal oceanic and Mediterranean pluviseasonal continental semicontinental upper mesomediterranean to lower supramediterranean lower subhumid to upper humid.

**T_avg_:** 6.3 **(8.2–11.5)** 16.4; **T_min_:** −3.1 **(−1.0–1.7)** 7.2; **T_max_:** 16.7 **(18.6–22.4)** 27.2; **P_avg_:** 276.5 **(574.2–1017.4)** 1572.6; **Pp:** 242.1 **(489.1–824.9)** 1227.5; **It:** 3.1 **(61.2–148.8)** 302.1; **Itc:** 8.5 **(66.2–171.9)** 311.0; **Tp:** 884.0 **(1046.0–1386.0)** 1977.0; **Ic:** 17.2 **(19.5–20.7)** 22.7; **Io:** 1.6 **(4.0–7.6)** 9.6; **Ios_2_:** 0.0 **(0.2–0.6)** 1.0; **Ios_3_:** 0.1 **(0.5–0.8)** 1.1; **Ios_4_:** 0.2 **(0.9–1.3)** 2.1.

#### 3.1.5. *Pinus balfouriana*

It appears associated with *Abies magnifica* A. Murray bis, *Pinus albicaulis* Engelm and *Pinus jeffreyi* A. Murray bis [81,122].


**Bioclimatic characterisation**


Mediterranean pluviseasonal oceanic and Mediterranean pluviseasonal continental strong semicontinental to subcontinental upper mesomediterranean to upper supramediterranean dry to subhumid.

**T_avg_:** 7.5 **(10.3–14.8)** 16.5; **T_min_:** −1.8 **(−1.1–4.1)** 6.9; **T_max_:** 18.1 **(21.4–26.5)** 28.2; **P_avg_:** 277.0 **(432.1–936.8)** 1301.3; **Pp:** 197.5 **(299.8–766.3)** 1072.4; **It:** 39.4 **(123.3–229.9)** 303.2; **Itc:** 44.0 **(136.9–269.2)** 316.2; **Tp:** 956.0 **(1257.0–1786.0)** 1975.0; **Ic:** 18.9 **(20.3–22.4)** 23.1; **Io:** 1.5 **(2.1–5.5)** 9.9; **Ios_2_:** 0.1 **(0.2–0.5)** 0.7; **Ios_3_:** 0.1 **(0.3–0.7)** 1.0; **Ios_4_:** 0.3 **(0.5–1.1)** 1.7.

#### 3.1.6. *Pinus jeffreyi*

This species is characteristic of the class *Calocedro decurrentis−Pinetea jeffreyi* Rivas−Martínez & Sánchez−Mata 1997; in particular, it occurs in all the associations of the order *Arctostaphylo patulae−Pinetalia jeffreyi* Rivas−Martínez & Sánchez−Mata 1997. The results of our research coincide with those proposed by these authors [119], but we would like to add that there are Temperate oceanic submediterranean localities for this species and that it also develops in hyperhumid ombrotypes.


**Bioclimatic characterisation**


Mediterranean pluviseasonal oceanic and Mediterranean pluviseasonal continental semicontinental upper mesomediterranean to upper supramediterranean upper dry to upper humid.

(Temperate oceanic submediterranean euoceanic upper supratemperate lower hyperhumid).

**T_avg_:** 4.7 **(9.1–14.4)** 20.7; **T_min_:** −4.8 **(0.0–6.4)** 12.7; **T_max_:** 15.2 **(18.7–24.0)** 32.0; **P_avg_:** 87.6 **(507.0–961.2)** 1941.3; **Pp:** 90.6 **(448.2–793.8)** 1663.2; **It:** −47.9 **(92.6–271.7)** 433.9; **Itc:** −41.8 **(100.6–274.1)** 437.1; **Tp:** 719.0 **(1108.0–1729.0)** 2484.0; **Ic:** 8.2 **(17.7–20.0)** 23.8; **Io:** 0.5 **(3.4–6.5)** 14.7; **Ios_2_:** 0.0 **(0.3–0.8)** 1.2; **Ios_3_:** 0.0 **(0.4–0.8)** 1.5; **Ios_4_:** 0.1 **(0.5–1.3)** 2.6.

#### 3.1.7. *Pinus longaeva*

Our results show that *P. longaeva* thrives in narrow intervals of bioclimatic values. This species is characteristic of the *Pinetum longaevae* association [119], i.e., Mediterranean xeric micro−forests whose peaks reach the oromediterranean upper semiarid. In our results, only mesomediterranean levels were reached.


**Bioclimatic characterisation**


Mediterranean xeric and Mediterranean desertic continental subcontinental upper mesomediterranean upper arid.

**T_avg_:** 11.8 **(22.9–11.8)** 22.9; **T_min_:** 0.9 **(3.0)** 9.4; **T_max_:** 23.4 **(26.0–26.1)** 36.6; **P_avg_:** 73.6 **(145.6–146.4)** 195.9; **Pp:** 73.5 **(139.5–139.6)** 168.4; **It:** 136.6 **(200.9–202.2)** 416.7; **Itc:** 175.0 **(245.15–247.67)** 529.7; **Tp:** 1456.0 **(1696.0–1702.0)** 2752.0; **Ic:** 22.5 **(23.1)** 27.3; **Io:** 0.3 **(0.8)** 1.1; **Ios_2_:** 0.1 **(0.1–0.2)** 0.4; **Ios_3_:** 0.1 **(0.1–0.2)** 0.4; **Ios_4_:** 0.3 **(0.2)** 0.5.

#### 3.1.8. *Sequoiadendron giganteum*

In the megaforest of *Corno nuttallii−Sequoiadendretum gigantei* Delgadillo in Rivas−Martínez & Sánchez−Mata 1997, this species is the dominant tree. This association occurs mainly in the upper supramediterranean and slightly in the lower oromediterranean and humid ombrotypes [119]. Our results are in agreement to a certain extent, since they do not show that the characteristic species (*S. giganteum*) of this association reaches the oromediterranean level.


**Bioclimatic characterisation**


Mediterranean pluviseasonal oceanic and Mediterranean pluviseasonal continental semicontinental upper mesomediterranean to lower supramediterranean upper dry to upper humid.

**T_avg_:** 7.9 **(11.3–14.9)** 17.3; **T_min_:** −1.5 **(2.2–5.8)** 9.7; **T_max_:** 18.0 **(21.2–25.3)** 28.0; **P_avg_:** 302.7 **(525.6–956.5)** 1393.1; **Pp:** 278.0 **(442.3–784.8)** 1362.2; **It:** 50.2 **(160.3–265.0)** 342.8; **Itc:** 56.2 **(166.0–276.2)** 343.5; **Tp:** 1005.0 **(1360.0–1794.0)** 2079.0; **Ic:** 9.5 **(18.6–20.1)** 21.5; **Io:** 1.8 **(2.9–6.3)** 11.3; **Ios_2_:** 0 **(0.1–0.4)** 1.1; **Ios_3_:** 0.1 **(0.2–0.5)** 0.9; **Ios_4_:** 0.2 **(0.5–1.0)** 1.3.

#### 3.1.9. Sequoia Sempervirens

This species is characteristic of the association *Lithocarpo densiflori−Sequoietum sempervirentis* Delgadillo in Rivas−Martínez & Sánchez−Mata 1997 [119]. The optimal characterisation of this association is defined by these authors as Mediterranean hyperoceanic supramediterranean humid to hyperhumid. This is partly consistent with our results, since the maximum value of hyperhumid was reached by this species in a few locations.


**Bioclimatic characterisation**


Mediterranean pluviseasonal oceanic subhyperoceanic to semihyperoceanic mesomediterranean subhumid to upper humid.

**T_avg_:** 8.4 **(12.6–14.9)** 18.8; **T_min_:** 0.3 **(7.6–9.0)** 12.8; **T_max_:** 15.4 **(17.5–20.7)** 25.7; **P_avg_:** 340.3 **(600.3–1092.7)** 1933.3; **Pp:** 340.1 **(608.5–1088.5)** 1907.8; **It:** 90.2 **(278.8–328.3)** 442.5; **Itc:** 92.6 **(274.4–324.6)** 444.9; **Tp:** 1029.0 **(1510.0–1786.0)** 2250.0; **Ic:** 7.8 **(9.8–12.5)** 19.0; **Io:** 1.8 **(3.6–6.8)** 14.7; **Ios_2_:** 0 **(0–0.2)** 0.8; **Ios_3_:** 0 **(0.1–0.4)** 1.5; **Ios_4_:** 0.1 **(0.3–0.9)** 2.7.

#### 3.1.10. Tsuga Mertensiana

This species is characteristic of the association *Phyllodoco breweri−Tsugetum mertensianae* Rivas−Martínez & Sánchez−Mata 1997. These authors [119] define it as oromediterranean, from humid to hyperhumid, in high areas where a large amount of snow accumulates. In our results, this species was essentially optimal in the upper supramediterranean and did not reach hyperhumid levels.


**Bioclimatic characterisation**


Mediterranean pluviseasonal oceanic and Mediterranean pluviseasonal continental semicontinental supramediterranean upper subhumid to upper humid.

**T_avg_:** 6.1 **(8.5–11.1)** 15.2; **T_min_:** −3.3 **(−0.7–1.8)** 5.7; **T_max_:** 16.6 **(18.9–22.3)** 26.6; **P_avg_:** 364.6 **(706.0–1091.4)** 1862.4; **Pp:** 274.7 **(612.1–920.3)** 1531.3; **It:** −4.5 **(74.4–144.5)** 264.8; **Itc:** 0.9 **(82.3–163.5)** 276.8; **Tp:** 860.0 **(1058.0–1354.0)** 1826.0; **Ic:** 13.5 **(19.4–20.4)** 22.4; **Io:** 2.1 **(4.9–8.1)** 13.9; **Ios_2_:** 0.1 **(0.3–0.6)** 1.0; **Ios_3_:** 0.2 **(0.5–0.8)** 1.4; **Ios_4_:** 0.3 **(1.0–1.4)** 2.4.

In short, all ten taxa studied occur in the Mediterranean macrobioclimate and, in particular, *Abies magnifica*, *Abies procera*, and *Pinus jeffeyi* can also thrive in a Temperate oceanic submediterranean macrobioclimate (Appendix A Table A2). The species that can be found in most of the isobioclimates are *Abies magnifica* and *Pinus jeffreyi*. These two taxa tolerate the greatest thermotypic range (from lower mesomediterranean to lower oromediterranean), and *P. jeffreyi* is the species that can thrive in a wide range of ombrotypes (from upper dry to lower hyperhumid). Of all the species studied, *Pinus longaeva* has the narrowest range of isobioclimates. *Picea sitchesis* and *Sequoia sempervirens* were also found to require the highest levels of oceanicity.

### 3.2. Habitat Suitability

Once the bioclimatic characterisation is known, this section explains the second aim of this study. Two types of figures are presented in this section. On the one hand, one type of figure (maps) shows the gains, losses, and maintenance of suitable and optimal areas for the development of each taxon studied until 2050, in relation to the current conditions (1980–2019). On the other hand, histograms (Figure 2) show the percentage of loss or gain of the currently suitable area for each of the species.

In short, by 2050, California is predicted to experience a decrease in ombrothermic index (Io) values (higher in the south than in the north) and an increase in the continentality index (Ic) values in the east. Widespread increases in both maximum and minimum temperatures throughout California and decreases in precipitation (except in the north and at high mountain elevations) are also observed, consistent with previous investigations [6,10] and IPCC projections [1,2,3]. The projections for the summer ombrothermic indices (Ios_2_, Ios_3_, and Ios_4_) have shown values that fall below one over the entire state, representing a significant decrease from the current values. Furthermore, the thermicity indices (It and Itc) have exhibited widespread increases projected by 2050, with the most notable increments observed in central and southern California. It is worth noting that the ten species studied are reliant on climatic conditions (temperature and precipitation) for their survival.

Analyses have revealed that climate change may lead to changes in the potential distribution of the taxa studied, as all of them will experience a decrease in suitable habitat in the future. The bioclimatic variables limiting the distribution of each species vary, but in general, increasing maximum temperatures, decreasing precipitation, and increasing summer drought are common limiting factors for species survival. A common distribution pattern for all species was observed as a shift from suitable areas to the north and/or higher elevations by 2050. The results for each species are presented in alphabetical order.

#### 3.2.1. *Abies magnifica*

*A. magnifica* currently occurs in the Sierra Nevada Province, the northern region of the Mendocino National Forest, and the northern part of the state (North and Central California province), as shown on the distribution map (Figure 3). The analysis of the results indicates that a significant portion of the species’ current suitable area (55.6%) will be unstable in the future (Figure 2).

By 2050, the northern suitable areas for *A. magnifica* will be at the higher elevations of the Sierra Nevada and around Mount Whitney. However, the current optimal areas in the Sierra Nevada and northern areas of Shasta Mountain will be unstable, leaving only a few optimal areas for the species. The results for 2050 show that the limiting bioclimatic variables for this species in these areas are T_max_, Ios_3_, and Ios_2_. Increasing maximum temperature, coupled with a general decrease in rainfall in the state [6], pose a threat to the survival of the northern populations of this tree [123]. This is consistent with previous research on trends in summer ombrothermic indices [124], which shows a negative trend in the indicated areas and the current habitat of this species [121]. The result is also in line with other research indicating a water balance deficit during the summer months for *A. magnifica* [33].

The map shows that the effects of climate change will have a negative impact on a part of the distribution of *A. magnifica* in California, which is consistent with other research showing the marked sensitivity and potential vulnerability of this tree to projected climatic changes, including warming trends and prolonged drought conditions [31].

#### 3.2.2. *Abies procera*

It is worth noting that *A. procera* has a very limited current range in northern California [54] in the western part of the Klamath National Forest (Figure 4), which is the Biogeographical Cascade Ranges and North and Central Provinces. The analysis of the variables has determined the suitable and optimal area for this species in the Sierra Nevada ranges, and we observed a higher suitable area than the one it currently occupies in the Klamath Mountains.

For this tree, we found that the northern areas it currently occupies will be threatened by the combined effect of changes in bioclimatic variables in 2050. Much of the suitable area will be adversely affected by decreases in positive precipitation (Pp) and increases in positive temperature (Tp) and T_max_ [123]. In addition, this species is particularly affected by the evolution of the continentality index (Ic). This index expresses the difference or thermal oscillation between the warmest and the coldest month of the year. In other words, the higher the value of Ic, and, therefore, the greater the temperature difference over the year, the more continental the area [124]. As described in the previous sections, this species is characterised by its occurrence with a strong influence of the ocean. It should be noted that the areas further away from the sea are more continental than those that are closer to it and are, therefore, attenuated by its influence. This is in line with previous research, which found a positive trend in the index in a large part of the state, with the exception of most coastal areas [124].

The analysis of the loss/gain of areas (as shown in Figure 2) reveals that the Noble fir species faces a significant increase in unstable suitable zones in the future, with 90% of the areas being affected. However, there is also a small gain of suitable area (5%) in the northernmost part of the state (Cascade Ranges Province) by 2050. Unfortunately, the areas currently occupied by the species are also in danger of being lost, with a loss of 6.6% of optimal area. The only areas that are expected to be stable or gained as optimal areas for the species are those adjacent to Jedediah Smith Redwoods State Park, in Cascade Range Province. These results are consistent with a recent study that identified Noble fir as the only species amongst the studied ones that did not show an increase in suitable habitat by 2080 [125].

#### 3.2.3. *Picea sitchensis*

Moving on to *P. sitchensis*, as can be seen from the map (Figure 5), the locations occupied by the species correspond to the suitable areas resulting from the analysis of the variables mentioned above. These areas are located in the Vancouver and Oregon Coast Sector and the North California coast district biogeographical regions. It thrives in waterlogged soils, but has limited tolerance to both extreme temperatures and frost and poor resilience to drought [126]. The results for 2050 indicate a certain threat to the survival of this species in the northwestern part of the state of California.

The summer ombrothermic indices Ios_2_ and Ios_3_ are the variables that have the greatest impact on the loss of suitable and optimal areas for this species. These indices not only determine the boundary between the Mediterranean and Temperate zones, but also help to identify summer drought. The observed reduction in the territories currently occupied by this species by 2050 pose a significant threat to its long−term survival, as it displaces its natural habitat. It is noteworthy that the future decline in the value of Ic may also affect some of the existing northernmost populations. Only a small northern suitable area would remain, corresponding to the areas adjacent to Crescent City in the Vancouver and Oregon Coast Sector. In this case, a large proportion of the optimal areas would be lost and a small percentage (Figure 2) of optimal areas (2%) would be gained in the north. This finding corroborates previous scientific research that has postulated the probable limitation of this species in the southern regions of Washington, Oregon, and California, while also suggesting its potential for northwestward expansion along the coastal areas of British Columbia, under projected environmental conditions [126].

#### 3.2.4. *Pinus albicaulis*

In the results for *P. albicaulis*, we found that the points of its current distribution are located mainly in the biogeographical province of the Sierra Nevada, south of Lake Tahoe, south of White Mountain (Figure 6). However, there is a potentially suitable area for the development of this tree in the surrounding areas of the San Bernardino National Forest. In the north of the state, it is mainly found to the west of Cedarville and in the surrounding areas of Mount Shasta and Klamath National Forest (North and Central California Province). Losses in these areas are mainly due to the change in the continentality index, especially the increase in values as found for these areas in other studies [124].

This forest will experience a high loss of suitable areas in the south of the state (60%) by 2050 (Figure 2), and only the highest points of the current sites would be stable in the future, thus reducing many suitable areas. These populations will be affected by the combined effect of several of the projected bioclimatic variables, particularly the increases in the values of T_max_ and Ic and the decrease observed in the values of Io, Ic, and Ios_3_ for 2050. However, the projections indicate that *P. albicaulis* is expected to retain stable suitable areas up to 40%. This finding is in line with other research highlighting the difference between high and low elevation systems. In fact, drought stress, as reflected by traditional climate variables and indices, may actually lead to a beneficial extension of the annual growing season. They propose that a longer growing season in these high−elevation systems promotes carbon accumulation and increased meristem activity, ultimately leading to increased tree growth and the development of protective structures [127].

In addition, a loss of the optimal ranges in the Sierra Nevada (Walker Mountain and Twin Peaks area) is predicted. This species would have optimal ranges around the Shasta−Trinity National Forest. It should be noted that there is no predicted increase in optimal range for this species in the future (2050). Comparable results were found in other research, with the greatest projected loss of whitebark pine by 2080 and 2100 in the Rocky Mountains [128].

#### 3.2.5. *Pinus balfouriana*

*P. balfouriana* is currently found in two disjunct populations in the state of California, one in the north in Klamath National Forest and the other in the central part of the state, in areas adjacent to Black Mountain, Inyo National Forest, and Mount Whitney (Figure 7). The results for this species show a large future loss of current suitable area in the southern Sierra Nevada Province, in addition to the disappearance of much of its optimal area. This area is the most affected by climate change in 2050 [124]. The convergence of decreasing mean and positive precipitation values, together with increases in T_avr_ and T_max_, has culminated in an upsurge in the continentality index, coupled with a reduction in the values of Ios_3_ [6,10,56,124]. In particular, it is the latter combination of factors that exerts the most inhibitory influence on the growth and development of this species. A small percentage (Figure 2) of suitable area is becoming more stable and gained to the north of the Sierra Nevada, particularly in the east of Yosemite National Park. Climate change is not expected to have a major negative impact on populations of this species in the north of the state, as suitable and optimal areas are also predicted to be stable and increased in these areas around Klamath Mountain. Other research showed that the existing populations of these trees may face challenges, not only in adapting to potential shifts in habitat suitability caused by abiotic conditions, but also in maintaining their presence in areas susceptible to pest infestations [129].

#### 3.2.6. *Pinus jeffreyi*

*P. jeffreyi* is distributed throughout the mountains of California (Figure 8). This tree can be found in the Southwestern California, the Sierra Nevada, and the North and Central California Provinces. Thus, we observe that the analysis of the variables reveals a large suitable area for the establishment of this species in the Central Valley Province. However, by 2050 we observe a loss of populations at low elevations and in the southern parts of the state. Thus, the predicted southern distribution of this species in 2050 in the south will be reduced to the San Bernardino National Forest, areas around Mount Palomar, and the higher elevations of Cleveland National Forest. Much of the current range in the southern Sierra Nevada will see its habitat restricted to higher elevations, where the species’ optimal range will be lost.

Based on our results, this species will have its distribution restricted mainly by the low values of Ios_2_ and Ios_3_ for 2050. It should be noted that these indices are directly proportional to the positive precipitation in the summer months and inversely proportional to the positive temperature in these months. The lower the value, the greater is the lack of water availability for plants in summer [48], especially for this species. We also note that an increase in suitable area is expected in the west and north of the state, as well as a gain of optimal area in the Klamath Mountains, where a large number of individuals in this species are currently found. For this species, a large proportion of suitable area is stable (40.2%) by 2050 (Figure 2). Given these results, the effects of climate change and projections for 2050 do not pose a major threat to the survival of this species in California, which is consistent with other research [33].

#### 3.2.7. *Pinus longaeva*

*P. longaeva* is restricted to eastern California (Figure 9), specifically to the areas around Inyo National Forest and Mount Witney [84,119,122]. The pooled analysis of the variables for this species includes a prediction of a very limited suitable area for the state. By 2050, a large loss of its current suitable area and optimal areas is expected. In the southern sites, the suitable area will be lost in 2050 due to the increase in temperature values (mean, maximum, and minimum). In the Mount Whitney sites, the bioclimatic variables that affect the development of this species are the decrease in positive precipitation values and in the summer ombrothermic indices Ios_2_ and Ios_3_. Increased drought by 2050, as defined by the latter two indices, is one of the main threats to the survival of this tree [130], although its physiology is highly adapted to drought [54,131]. This tree will spread to a small area of suitable land to the east of the current locations in the Intermountain Province (Figure 2). Given our results on the range of this species, it is likely that it will disappear from the state of California.

#### 3.2.8. *Sequoiadendron giganteum*

The geographic distribution of *S. giganteum* in California is currently restricted to the southern Sierra Nevada Province, specifically in the Sequoia National Forest. However, it is worth noting that its occurrence is limited to very restricted areas in the northwestern regions of Los Angeles and San Bernardino. There are a few specimens in eastern Sacramento and a few sites in Lassen National Forest (Figure 10). Areas occupied by this species are characterised by adequate soil moisture in summer, moderate temperatures in winter, humidity, and frequent fires [132]. The projected outcomes illustrated on the map indicate that by 2050, this species will experience a significant loss of its current suitable habitat, equivalent to 79% (Figure 2). Furthermore, the areas of gain and retention will be significantly reduced to 38%, mainly at the higher elevations of its existing range. However, there is a positive trend of a 21% gain in suitable habitat towards the northern regions of the state, as well as certain areas in the Sierra Nevada. This is due to an increase in one of the factors that determines its natural habitat, summer drought [34,45,93].

As we have already seen in the southern half of the state, the values of the summer ombrothermic indices Ios_2_ and Ios_3_ will decrease by 2050. This negative trend has been observed in other research for California [124]. Another factor distressing the habitat of this species in the south is the increase in Ic, i.e., an increase in the thermal amplitude which, as we have already mentioned, is unfavourable for the development of this tree.

It should be noted that *S. giganteum* will lose the optimal areas it currently occupies in the northern parts of the Sierra Nevada, gaining optimal areas (1%) in the south of Shasta Trinity National Forest and at the highest elevations of the current suitable areas (Figure 10).

#### 3.2.9. Sequoia Sempervirens

*S. sempervirens* currently occupies the west coast of California in the most oceanic parts (Figure 11), where dense summer fog provides moisture to the trees during the dry months [133]. Note that there are some current locations in the Sacramento Valley. The analysis of the results predicts a large loss of the suitable area in the southern parts of the state, with the western zone of San José remaining suitable and the current areas of southern Monterrey being lost. This is due to decreasing values of the annual ombrothermic index (Io) by 2050, with very low values (dry) in the southern half of the territory, and to declining values of Ios_3_.

In addition, optimal losses for this species are expected to occur in the north of San Francisco and surrounding areas of Mendocino by 2050. In the most oceanic areas, there is an increase in optimal areas, as well as a gain in suitable areas upstate in the Eureka and Crescent city surroundings (Cascade Ranges Province). There will be a reduction of 76% of the suitable area as opposed to a gain of 1.75% (Figure 2). The survival of this forest is variable, but this could be explained by metrics related to site water balance [134], such as considering summer ombrothermic indices.

#### 3.2.10. Tsuga Mertensiana

*T. mertensiana* is located in the Sierra Nevada in the mountains of Yosemite National Park and near Tahoe National Forest (Figure 12). It is also found in the surrounding areas of Yreka and the Klamath National Forests, as well as in the northernmost part of the state in the Pacific Coast Range, usually on cold and snowy sites [54]. The results obtained for this species predict a large loss of suitable areas (71.7%) by 2050 across all the lower elevations in southern sites of the Sierra Nevada and Klamath Mountains (Figure 2). This is in line with previous research, which found that amongst topographic variables, elevation appears to be the one that most influences the southerly distribution of mountain hemlock [135]. This implies, as shown in the map, the disappearance of optimal areas for this species. Considering the bioclimatic variables of our study, it is observed that the occurrence of this species in 2050 in the southern part of the Sierra Nevada is limited by a decrease in mean precipitation and positive precipitation [10,124] together with an increase in continentality. However, in the northern parts of this mountain system, area loss is driven by decreasing values of Ios_3_. Further north, in the Klamath areas, the main variable inhibiting the development of this species is the increase in Ic values. However, there is a gain in suitable areas (7%) towards the north–west of the state (Cascade Ranges Province), maintaining a large proportion of sites (25%) in the Klamath Mountains. In addition, the areas surrounding Fort Jones and Callahan were found to be optimal. Although our study has identified both increases and decreases in suitable areas for these species, other research has shown only declines in future projections [128].

By and large, most of the native conifers studied here will endure a reduction in their habitat range in California by 2050. This is consistent with previous research that has found high tree mortality due to climatic change as well as the displacement of individuals to optimal areas [34,35,45,136], where current forest cover will be difficult to reach due to the additional effects of interspecific competition and fire [70,137,138,139]. In addition, the results have brought to the fore the applicability of bioclimatology under future conditions of climate change.

## 4. Conclusions

This research has provided for the first time both the quantitative and qualitative bioclimatic characterisation of ten native conifers in California. We have numerically determined the maximum, minimum, and interquartile range values of a large number of indices and bioclimatic parameters in which the ten conifer taxa studied occur. In addition, we have extended the previously established definitions for the qualitative characterisation (isobioclimate) of the taxa. Furthermore, the focal objective of this study is to examine the habitat suitability for the species under examination after carrying out a comprehensive analysis. The study is primarily concerned with providing cartographic depictions of habitat suitability, coupled with histograms illustrating the percentage loss, gain, and maintenance of the suitable areas for each species under climate change projections. Moreover, this study emphasises the paramount role that bioclimatology plays in the survival of the species studied. The following conclusions can be drawn from the results obtained:In the qualitative bioclimatic characterisation, we have observed that *Abies magnifica*, *Abies procera*, and *Pinus jeffreyi* occur in some localities with a Temperate submediterranean macrobioclimate, while the rest of the species develop under a Mediterranean macrobioclimate. Many of the qualitative diagnoses of the species studied are consistent with previous studies, while this research has qualified some of the existing ones.Through the projection of future bioclimatic conditions, a compelling revelation emerges: California is poised to experience a significant decline in annual ombrothermic index (Io) values, contrasting with a remarkable surge in continentality index (Ic) by the year 2050. Furthermore, widespread temperature increases and precipitation reductions are anticipated, with few exceptions in the north and elevated mountain regions. From a broader perspective, the environmental factors, parameters, and bioclimatic indices that exert the greatest impact on the conifers studied encompass increasing continentality index values, indicative of annual thermal amplitude, as well as declining summer ombrothermic indices Ios_2_ and Ios_3_, indicative of increasing summer drought. The overall trend observed for suitable and optimal habitats for these species in the future is a shift towards northern regions and higher altitudes.*Abies procera* and *Pinus longaeva* are the species poised to lose most of their suitable areas (90%) in the future (2050).The conifer species that are projected to retain the largest proportion of suitable habitat in the future, specifically by 2050, are *Abies magnifica*, with 43% of its current range, and *Pinus albicaulis*, with nearly half of its present distribution, namely 40%.It is abundantly clear that the loss of suitable areas for the *Sequoiadendron giganteum* is due to the increase in summer drought, which is one of the factors determining its natural habitat. In particular, the intensification of summer droughts will render 79% of its suitable areas unsuitable by mid−century.Most of the conifers studied here will endure a reduction in their habitat range in California by 2050.Bioclimatology proved to be a relevant approach to understand the ecological responses to changing environmental conditions due to climate change.

We consider that the findings of this research provide compelling evidence for the use of bioclimatic models to predict the future of species ranges under climate change scenarios, and the potential for their application in conservation planning. Indeed, the ecological consequences of conifer loss are not limited to the forest ecosystem, but may have far−reaching implications given the ecosystem services they provide, including carbon sequestration, water regulation, and biodiversity conservation.

## Figures and Tables

**Figure 1 plants-12-01966-f001:**
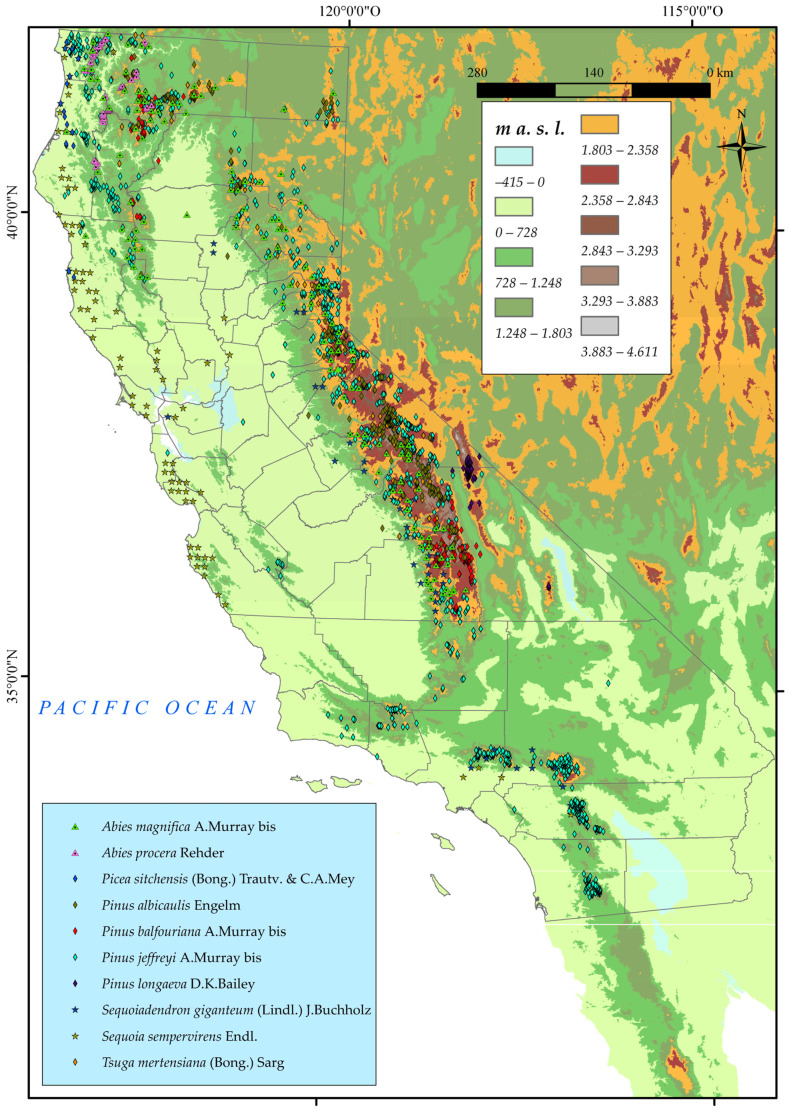
Distribution of the ten chosen taxa (Pinales Gorozh) in California.

**Figure 2 plants-12-01966-f002:**
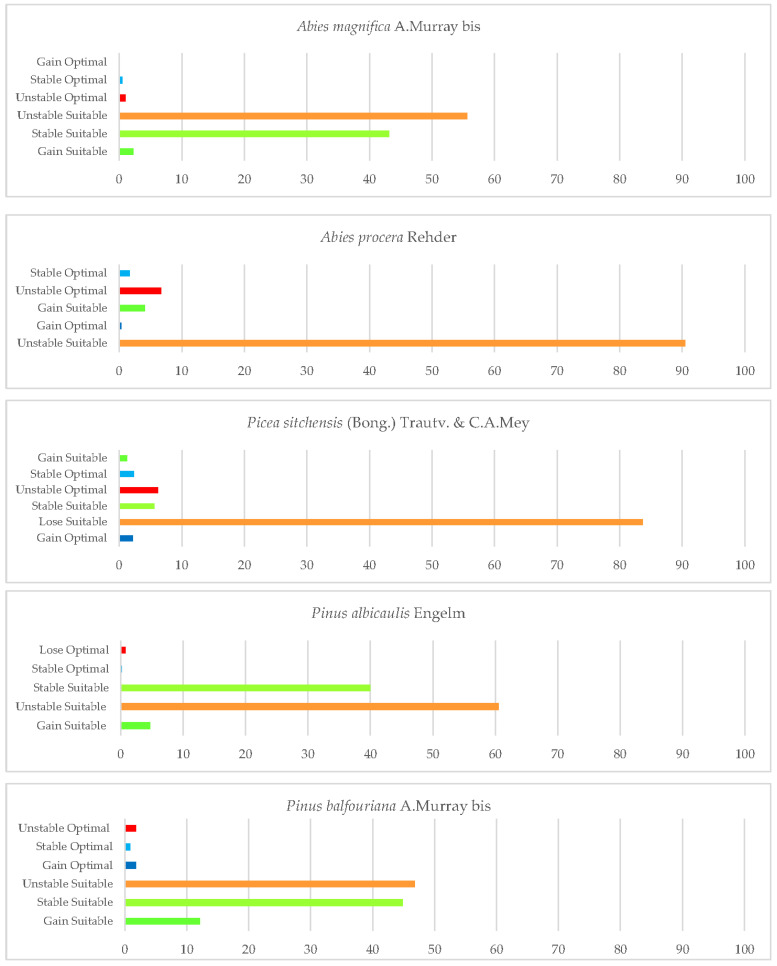
Percentages of gain, unstable, and stable suitable and optimal areas of the studied Pinales between the reference period (1980–2019) and the future (2050).

**Figure 3 plants-12-01966-f003:**
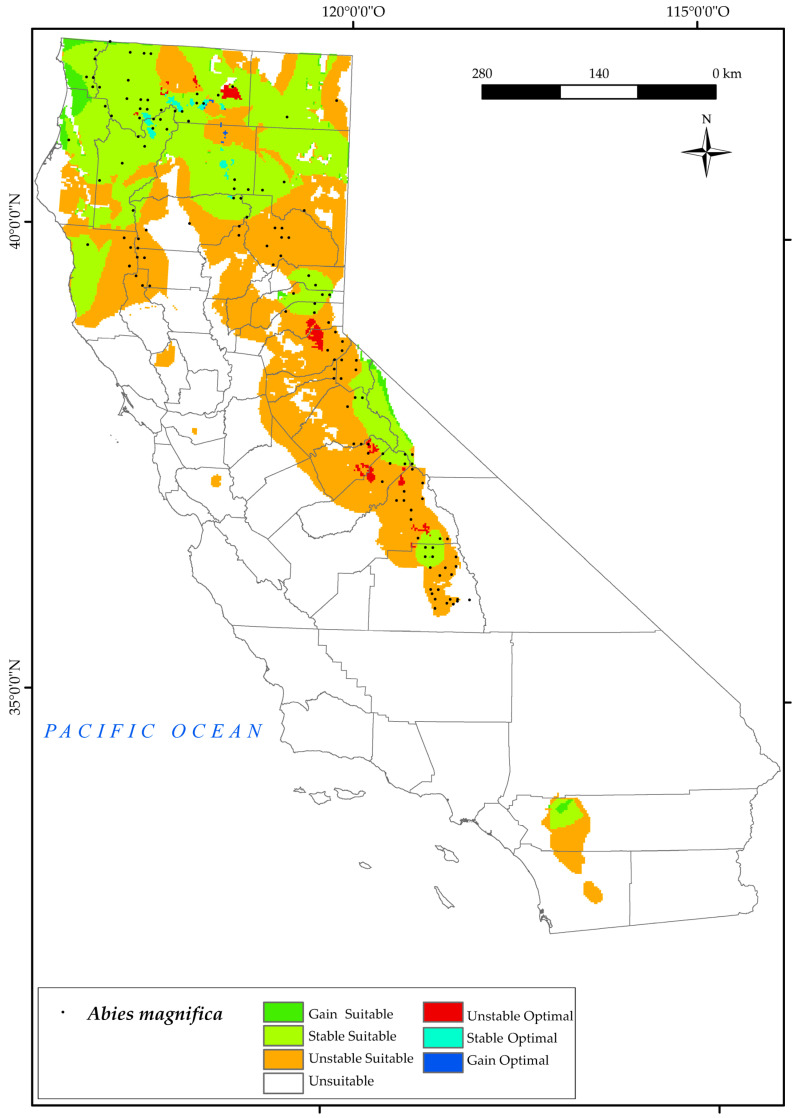
Variation in suitable and optimal areas of *Abies magnifica* between the reference period (1980–2019) and the future (2050). (●) Species presence sites.

**Figure 4 plants-12-01966-f004:**
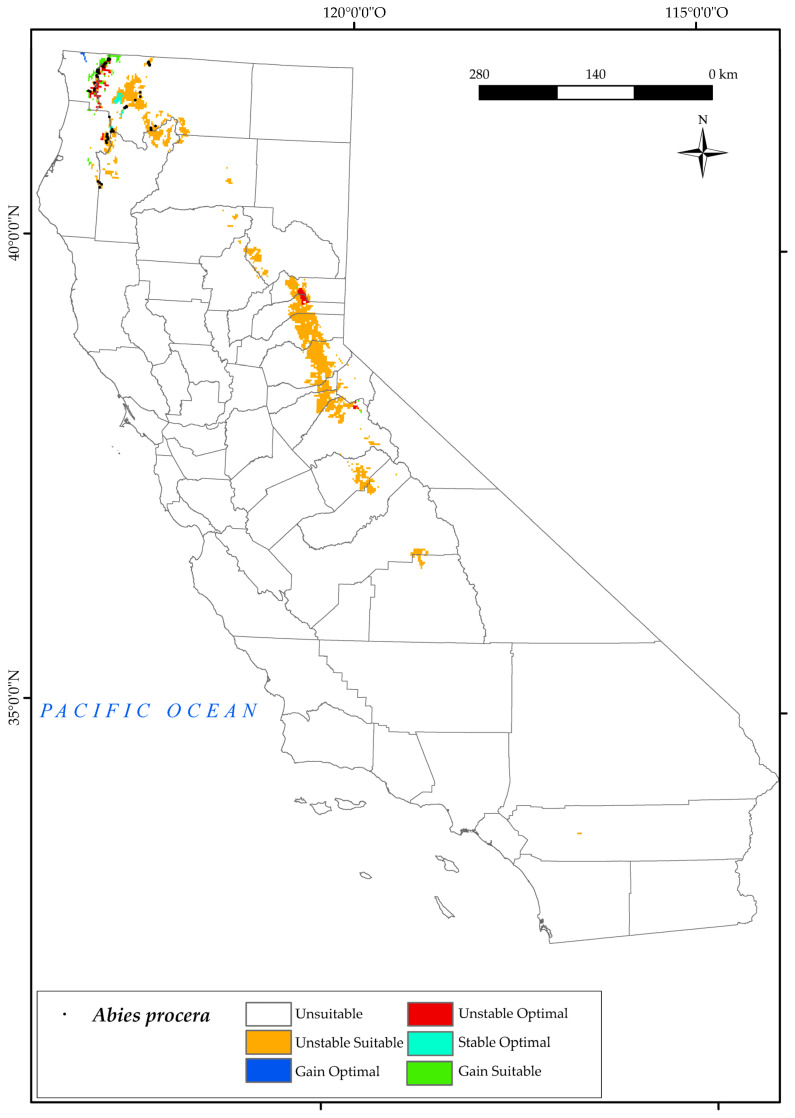
Variation in suitable and optimal areas of *Abies procera* between the reference period (1980–2019) and the future (2050). (●) Species presence sites.

**Figure 5 plants-12-01966-f005:**
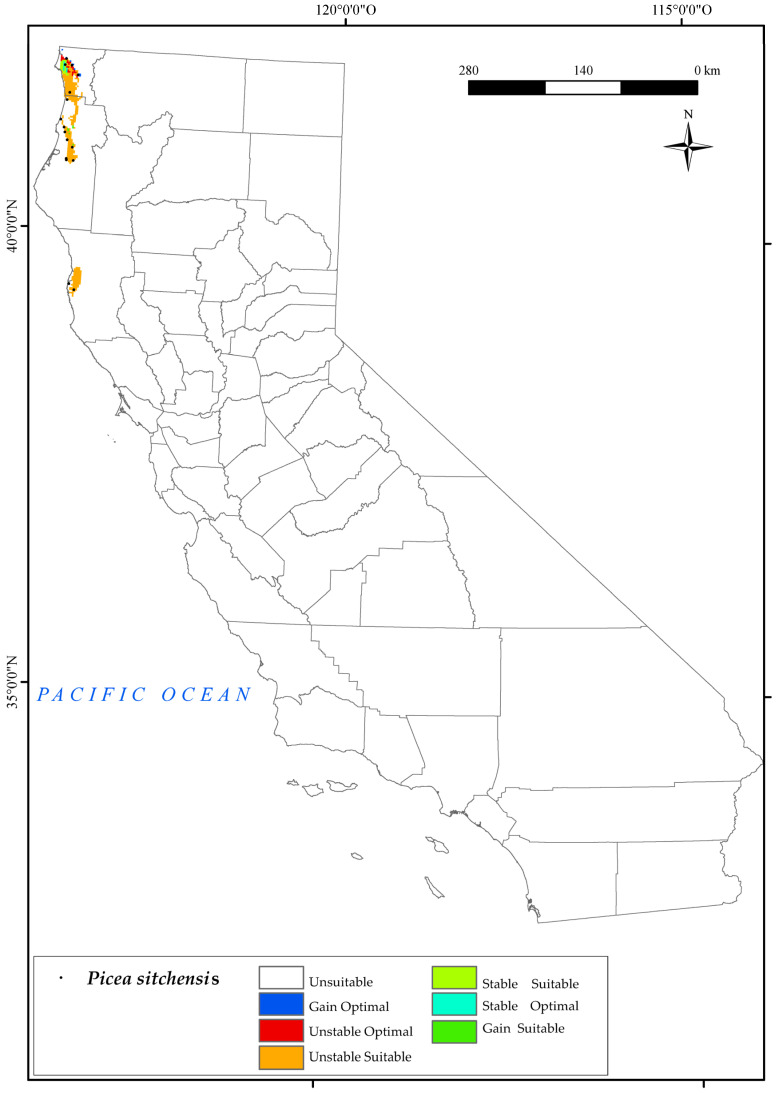
Variation in suitable and optimal areas of *Picea sitchensis* between the reference period (1980–2019) and the future (2050). (●) Species presence sites.

**Figure 6 plants-12-01966-f006:**
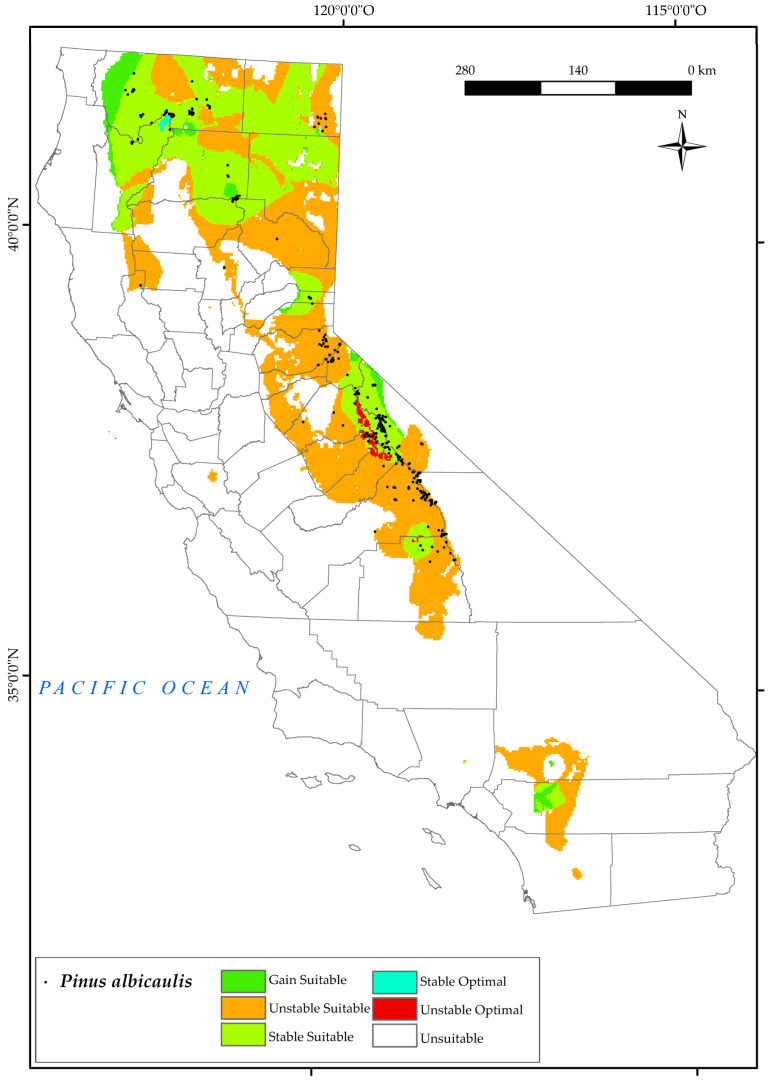
Variation in suitable and optimal areas of *Pinus albicaulis* between the reference period (1980–2019) and the future (2050). (●) Species presence sites.

**Figure 7 plants-12-01966-f007:**
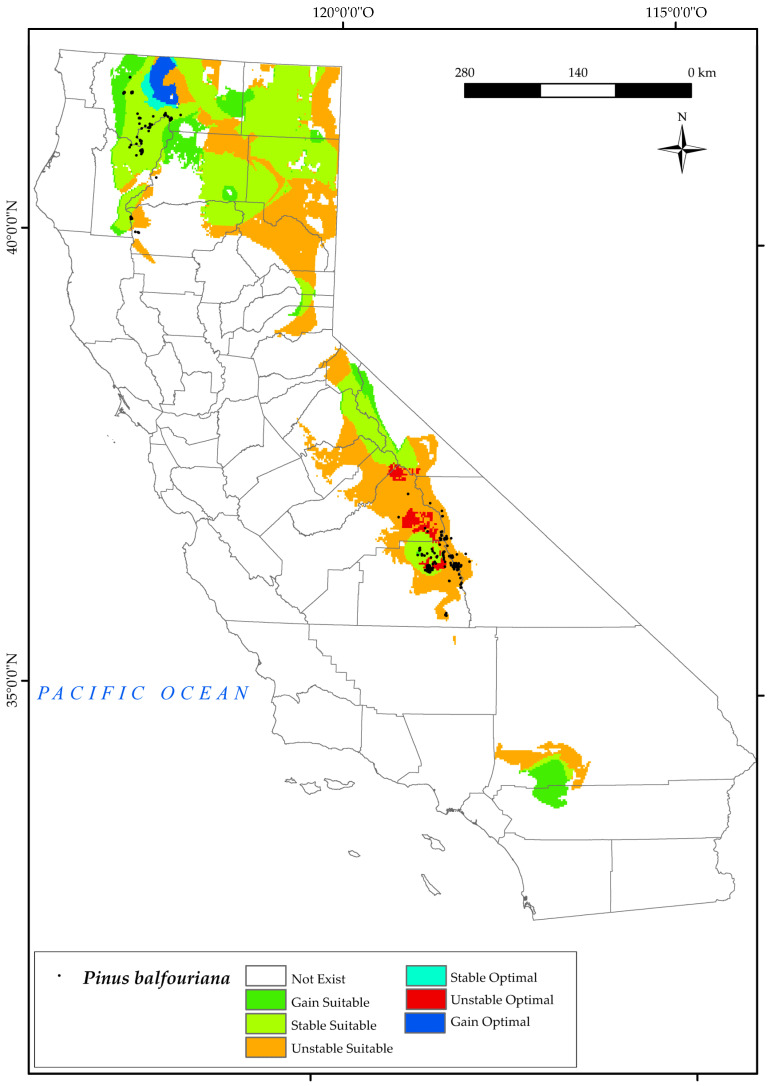
Variation in suitable and optimal areas of *Pinus balfouriana* between the reference period (1980–2019) and the future (2050). (●) Species presence sites.

**Figure 8 plants-12-01966-f008:**
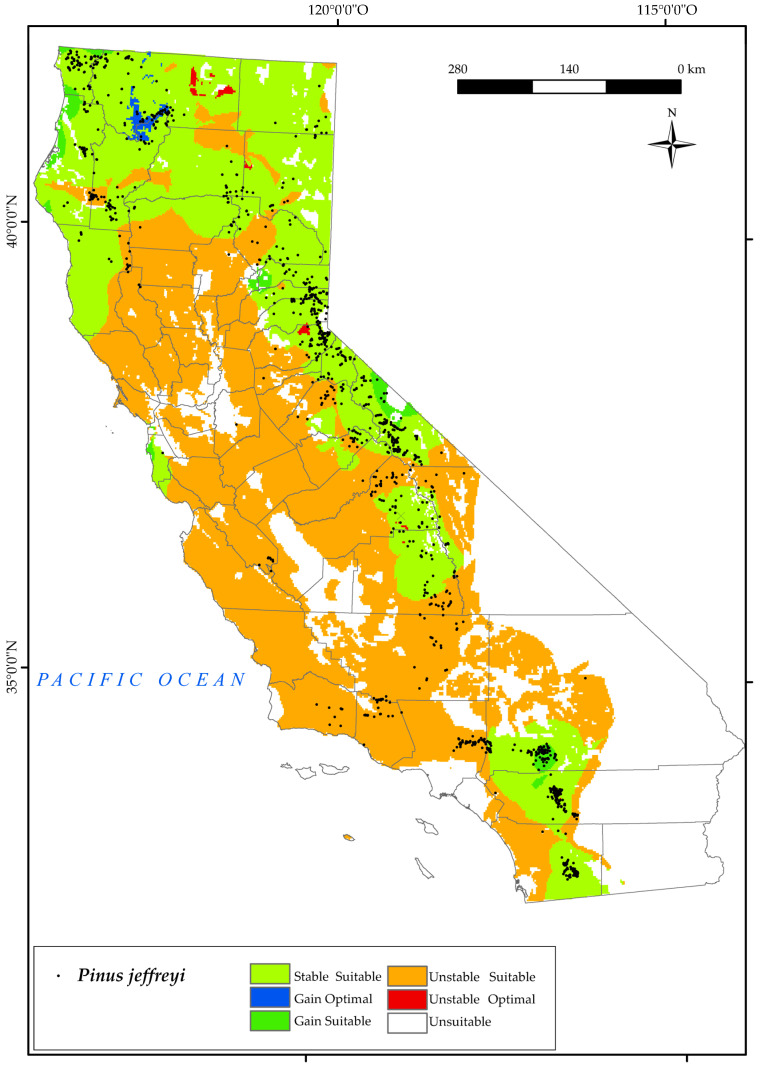
Variation in suitable and optimal areas of *Pinus jeffreyi* between the reference period (1980–2019) and the future (2050). (●) Species presence sites.

**Figure 9 plants-12-01966-f009:**
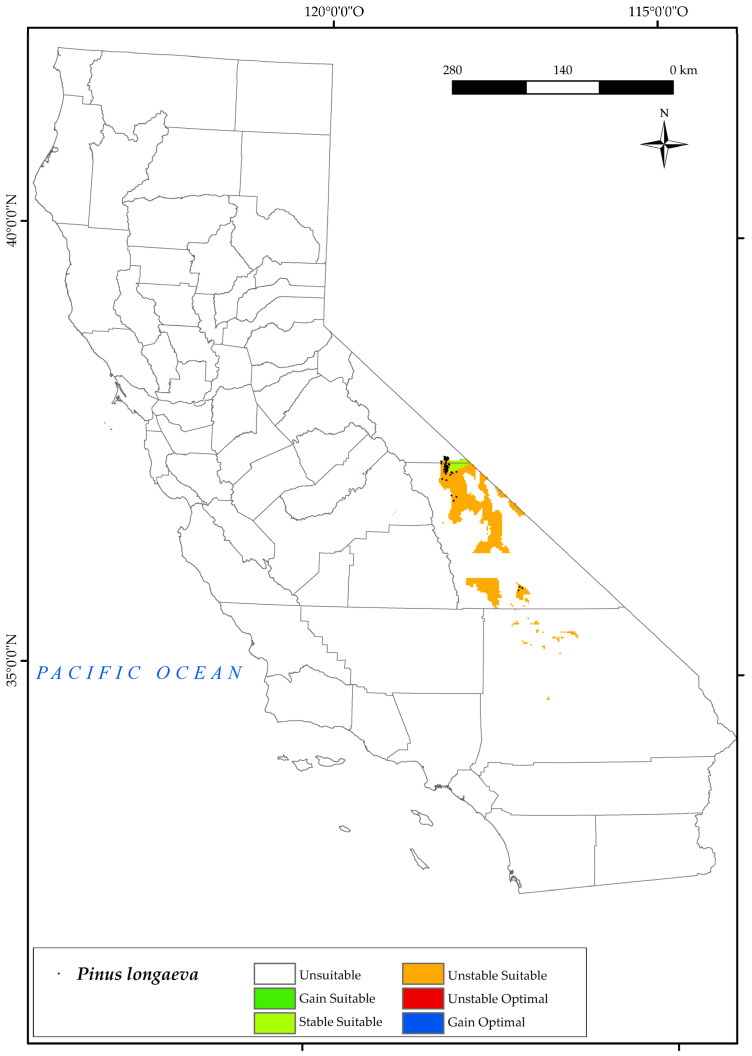
Variation in suitable and optimal areas of *Pinus longaeva* between the reference period (1980–2019) and the future (2050). (●) Species presence sites.

**Figure 10 plants-12-01966-f010:**
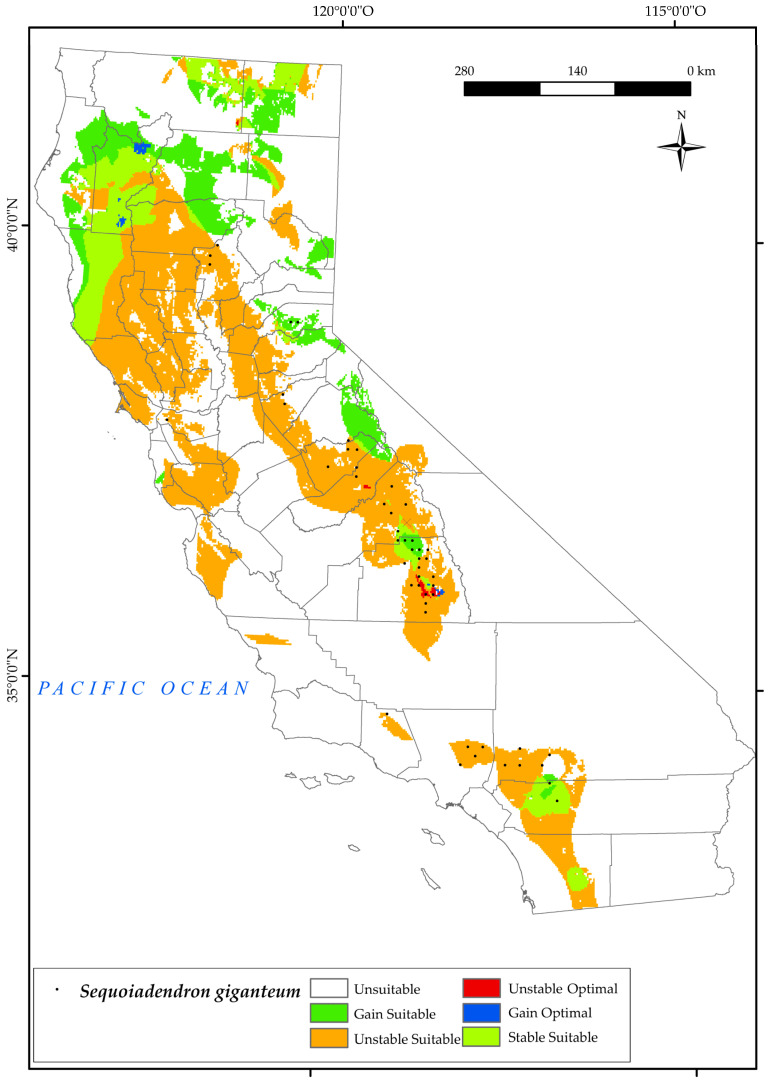
Variation in suitable and optimal areas of *Sequoiadendron giganteum* between the reference period (1980–2019) and the future (2050). (●) Species presence sites.

**Figure 11 plants-12-01966-f011:**
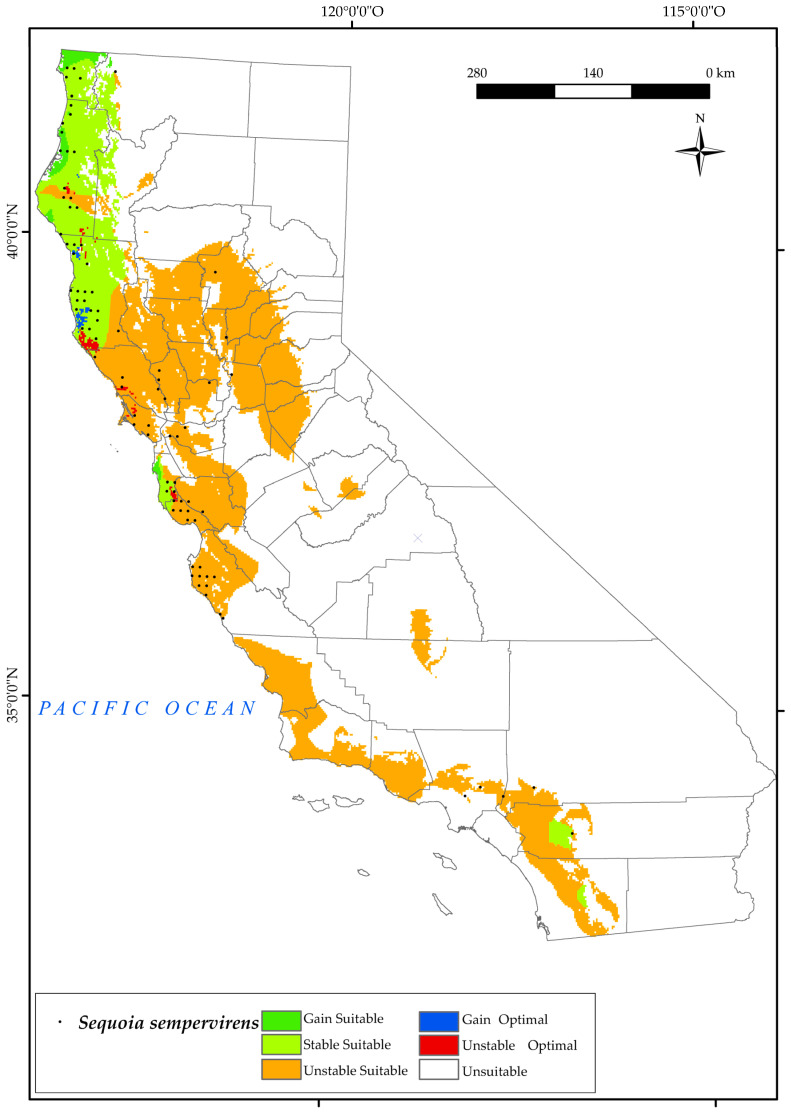
Variation in suitable and optimal areas of *Sequoia sempervirens* between the reference period (1980–2019) and the future (2050). (●) Species presence sites.

**Figure 12 plants-12-01966-f012:**
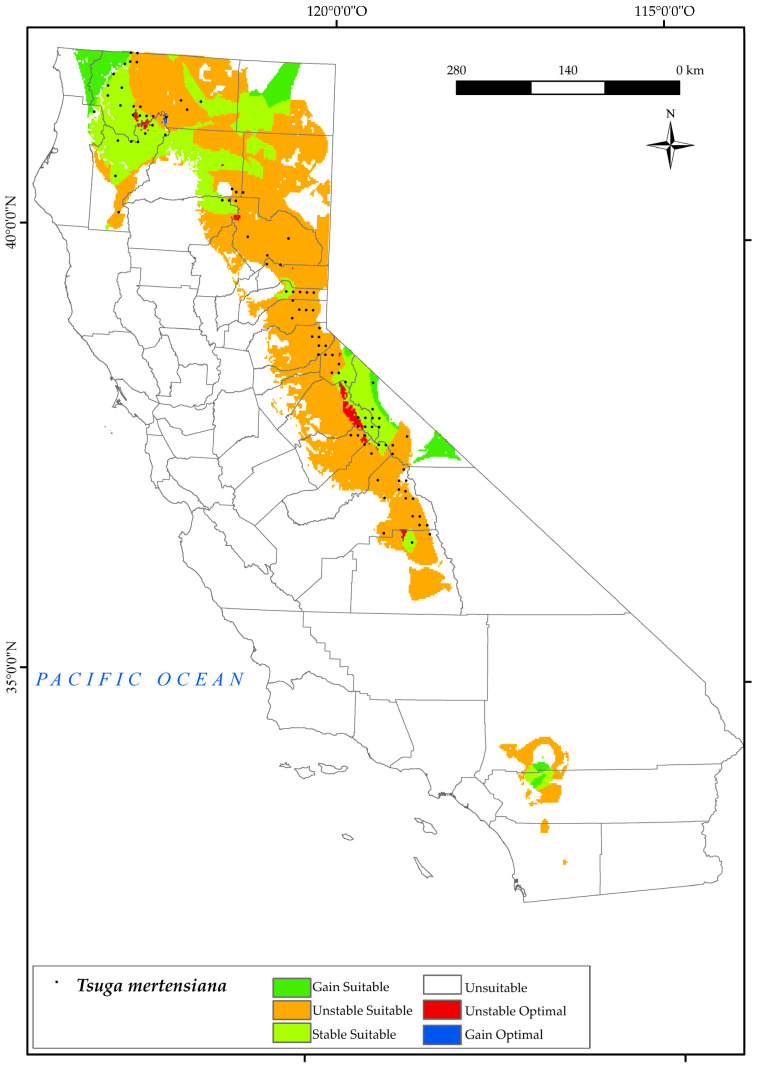
Variation in suitable and optimal areas of *Tsuga mertensiana* between the reference period (1980–2019) and the future (2050). (●) Species presence sites.

**Table 1 plants-12-01966-t001:** Bioclimatic parameters and indices applied in this study. Taken from Rivas−Martínez et al. 2011.

Index Name	Definition
Average annual Temperature (Tavr)	Average annual temperature in degrees Celsius.
Average annual Precipitation (Pavr)	Annual precipitation in millimetres.
Positive annual Temperature (Tp)	Sum of the temperatures of the months whose average temperature is greater than 0 °C. It is expressed in tenths of a degree.
Maximum Temperature (T_max_)	Average temperature of the hottest month of the year.
Minimum Temperature (T_min_)	Average temperature of the coldest month of the year.
Positive annual Precipitation (Pp)	Sum of the average precipitation in millimetres of the months whose average temperature is greater than 0 °C.
Simple Continentality Index (Ic)	Difference or oscillation between the mean temperature of the warmest month (T_max_) and that of the coldest month of the year (T_min_). Ic = T_max_ − T_min_.
Thermicity Index (It)	It can be calculated as the average annual temperature plus twice the temperature of the coldest month, and all this multiplied by ten. It is, therefore, an index that values the intensity of the cold.
Compensated Thermicity Index (Itc)	Index that attempts to weight the value of the thermicity index (It) due to the “excess” of cold or temperance that occurs during the cold season in the territories of marked continental or hyperoceanic climate on Earth. In addition, this Index provides thermotype characterisation.
Annual Ombrothermic Index (Io)	This index is the quotient between the positive precipitation (Pp) and the positive temperature (Tp) multiplied by ten.
Summer Ombrothermic Indices (Ios_i_)	Ios_1_ (ombrothermic index of the warmest month of the summer quarter); Ios_2_ (ombrothermic index of the hottest two months of the summer quarter); Ios_3_ (ombrothermic index of the summer quarter); and Ios_4_ (calculated with the months of the summer quarter and the previous month).

**Table 2 plants-12-01966-t002:** Example of bioclimatic units that structure the qualitative bioclimatic characterisation for a taxon.

Macrobioclimate	Bioclimate (Variants)	Continentality	Thermotype (Horizons)	Ombrotype (Horizons)
**Mediterranean**	Pluviseasonal oceanic	Semicontinental	Lower mesomediterranean	Upper subhumid

## Data Availability

The datasets generated and analysed during the current study are not publicly available due to the fact that the R package is in the process of being published. However, the original data source can be consulted at https://wrcc.dri.edu/ (accessed on 8 May 2023) and are available from the corresponding author upon reasonable request.

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
