# Peer review of "Bioclimatic Characterisation of Specific Native Californian Pinales and Their Future Suitability under Climate Change"

_plants, 2023, doi:10.3390/plants12101966_

Round 1
Reviewer 1 Report
The article needs improvement in some aspects.
In the methodology ,it's stated that only the results are shown (L275). I think it would be convenient to show the results obtained in the other methods as a justification of choice.
I have not been able to locate in the text which scenario or future category has been used.
In all species the climatic variables are shown at the end. I think that Table 1 of Annex I is sufficient. It's clearer than in the text, so my opinion is to remove the text for each species, explaining that these data appear in Annex I.
The images of the percentages, from my point of view, should have the same scale, as the size of the bars can be misleading.
To avoid having so many images, could all the percentage images be unified into a single image?
I think that with these changes the document would improve in its compression.
Reviewer 2 Report
This article about the bioclimatic characterization for Mediterranean distributed conifers, is very important, since it brings novelty to a necessary and up-to-dated analysis. It is particularly relevant and interesting, since it enhances geobotanical analysis in the framework of the renewed ecological sciences, bringing new insights and complementing the conventional methods normally used in niche forecast. For me this is the most important achievement of this work, since it brings awareness to the models developed by the Worldwide Bioclimatic Classification System by Rivas-Martínez et al. (2011), that suffers from lack of attention from the scientific community.
The Introduction is not particularly “crisp”, and it is extensive. It would be advisable to reduce some unnecessary information about nature-based solutions and other themes inserted “out of the blue”, that in the end are neither used nor discussed downstream. Instead, I have missed the biogeographic importance of California for the worldwide Mediterranean Biome, and it would benefit if a specific group of species were enhanced, also from the phylogeographic and ecological point of view. It also lacks a global overview of the landscape positioning of conifers in face of angiosperms, namely oaks (Quercus), since these conifers broadly constitute edaphoxerophile series and are commonly related with depleted soils (especially Pinus) and all the importance of the ecological positioning of these group of species in the modelling framework. They may be related with areas with different bioclimatic needs, but they are also heavily conditioned by geology, soils and topography.
The chosen methods and climatic analysis are correct and well standardized and delineated.
Authors should privilege the terminology climatic suitability, instead of habitat range/suitability, since they are only operating with climate and excluding edaphic variables.
The English is understandable, but the entire manuscript would benefit from a native speaker revision.
Results are very complete; however, I would remove the species information for supplementary information (Focal Taxa), and in all species, the final paragraphs discussion about the predicted changes could be summarized in an own section. These can be solved by providing “quicker” results for all species and provide a subsection called “Species climatic range shifts” where the authors could broadly discuss all parameters for each species or group of species with similar trends, in the light of the obtained results.
The maps are quite informative and the major output of the article. I suggest that the authors reframe the terminology as: Stable suitable instead of “Keep” suitable, “unsuitable” for Not exist, and Unstable Suitable and Unstable Optimal instead of “Lose” Suitable and “Lose” optimal.
The authors should substitute the terminology “develop” when referring to species existence to “occurr” or “occurrence”.
I recommend that the article could be published in Plants, under the necessary amendments, that I leave in the PDF revised file.

The English would benefit a full review by a native speaker, since the authors tend to write in a Romantic manner.
Round 2
Reviewer 1 Report
Thank you for making the changes.
The document is ready to be published.